# STATE ADVANTAGE WEIGHTING FOR OFFLINE RL

**Jiafei Lyu**[1,*] **Aicheng Gong**[2], **Le Wan**[3], **Zongqing Lu**[4], **Xiu Li**[1]

[1]Tsinghua Shenzhen International Graduate School, [2]China Nuclear Power Engineering Company Ltd
[3]IEG, Tencent, [4]School of Computer Science, Peking University

## ABSTRACT

We present *state advantage weighting* for offline reinforcement learning (RL). In contrast to action advantage $A(s, a)$ that we commonly adopt in QSA learning, we leverage state advantage $A(s, s')$ and QSS learning for offline RL, hence decoupling the action from values. We expect the agent can get to the high-reward state and the action is determined by how the agent can get to that corresponding state. Experiments on D4RL datasets show that our proposed method can achieve remarkable performance against the common baselines. Our code is publicly available in https://github.com/dmksjfl/SAW.

## 1 INTRODUCTION

In this paper, we explore a novel QSS-style learning paradigm for offline RL Lange et al. (2012). We estimate the state $Q$-function $Q(s, s')$, which represents the value of transitioning from the state $s$ to the next state $s'$ and acting optimally thenceforth: $Q(s, s') = r(s, s') + \gamma \max_{s'' \in \mathcal{S}} Q(s', s'')$. By doing so, we decouple actions from the value learning, and the action is determined by how the agent can reach the next state $s'$. The source of OOD (out-of-distribution) will then turn from next action $a'$ into next next state $s''$. In order to get $s''$, we additionally train a predictive model that predicts the feasible and high-value state. We deem that this formulation is more close to the decision-making of humans, e.g., we predict where we can go and then decide how we can get there when climbing.

Unfortunately, we find that directly applying D3G Edwards et al. (2020), a typical QSS-learning algorithm, is infeasible in offline settings. We wonder: *can QSS learning work for offline RL?* Motivated by IQL Kostrikov et al. (2022), we propose to learn the value function by expectile regression Koenker & Hallock (2001) such that both the state $Q$-function $Q(s, s')$ and value function $V(s)$ can be well-trained. We train extra dynamics models for predicting the next next state $s''$. We train an *inverse dynamics model* $I(s, s')$ to determine the action, i.e., how to reach $s'$ from $s$. We leverage *state advantage* $A(s, s') = Q(s, s') - V(s)$, which describes how the next state $s'$ is better than the mean value, for weighting the update of the actor and the model. To this end, we propose **S**tate **A**dvantage **W**eighting (SAW) algorithm. We conduct numerous experiments on the D4RL benchmarks. The experimental results indicate that SAW is competitive or even better than the prior methods.

## 2 PRELIMINARIES

In QSA learning, the action $Q$-function defines the expected discounted return by taking action $a$ in state $s$ under policy $\pi$: $Q^\pi(s, a) = \mathbb{E}_\pi[\sum_{t=0}^\infty \gamma^t r_{t+1} | s_0 = s, a_0 = a]$. The action advantage is defined as: $A(s, a) = Q(s, a) - V(s)$, where $V(s)$ is the value function. The Q-learning gives: $Q(s, a) \leftarrow Q(s, a) + \alpha[r + \gamma \max_{a' \in \mathcal{A}} Q(s', a') - Q(s, a)]$. The action is then decided by $\arg\max_{a \in \mathcal{A}} Q(s, a)$. In QSS learning, we focus on the state $Q$-function: $Q(s, s')$. That is, the value in QSS is independent of actions. The action is determined by an inverse dynamics model $a = I(s, s')$, i.e., what actions the agent takes such that it can reach $s'$ from $s$, $\pi : \mathcal{S} \times \mathcal{S} \mapsto \mathcal{A}$. We can similarly define that the optimal value satisfies $Q^*(s, s') = r(s, s') + \gamma \max_{s'' \in \mathcal{S}} Q^*(s', s'')$. The Bellman update for QSS gives Edwards et al. (2020): $Q(s, s') \leftarrow Q(s, s') + \alpha[r + \gamma \max_{s'' \in \mathcal{S}} Q(s', s'') - Q(s, s')]$. *State advantage* $A(s, s') = Q(s, s') - V(s)$ measures how good the next state $s'$ is over the mean value.

---

*Work done while working as an intern at Tencent IEG.

## 3 STATE ADVANTAGE WEIGHTING

D3G is a typical QSS learning algorithm. It aims at learning a policy under deterministic transition dynamics. Unfortunately, D3G exhibits very poor performance on continuous control tasks with its vanilla formulation (e.g., Walker2d-v2). Will D3G succeed in offline settings? We examine this by conducting experiments on hopper-medium-v2 from D4RL Fu et al. (2020) MuJoCo datasets. We observe in Figure 1(a) that D3G fails to learn a meaningful policy on this dataset. As shown in Figure 1(b), the $Q$ value (i.e., $Q(s, s')$) is extremely overestimated (up to the scale of $10^{12}$). We wonder *can we make QSS learning work in offline RL?* This is vital due to its potential for promoting learning from observation and goal-conditioned RL in the offline manner. To this end, we propose **S**tate **A**dvantage **W**eighting (SAW). We observe that SAW exhibits very good performance on hopper-medium-v2, with its value estimated fairly well, as shown in Figure 1(c).

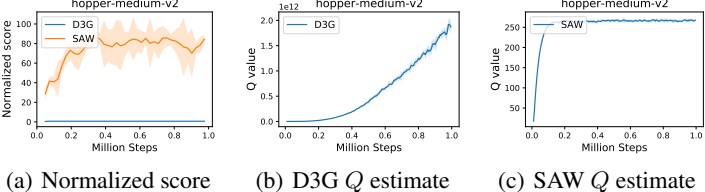

| (a) Normalized score | (b) D3G $Q$ estimate | (c) SAW $Q$ estimate |

Figure 1: Normalized score comparison of D3G against SAW on hopper-medium-v2 from D4RL (a). The $Q$ value estimate of D3G incurs severe overestimation (b) while our SAW does not (c). The results are obtained over 5 random runs, and the shaded region captures the standard deviation.

The SAW is consisted of five components: the state $Q$-function $Q(s, s')$, the forward model $F(s, a)$ (to ensure consistency), the inverse dynamics model $I(s, s')$, the value function $V(s)$, and the prediction model $M(s)$ (to predict $s'$). The details of training SAW can be found in Appendix C.2.

For experimental evaluation, we compare our SAW algorithm against some common baselines in offline RL, CQL Kumar et al. (2020), behavior cloning (BC), Decision Transformer (DT) Chen et al. (2021), UWAC Wu et al. (2021), TD3+BC Fujimoto & Gu (2021), and IQL. All algorithms are run over 5 different random seeds. We present the comparison results in Table 1, where we observe that SAW shows competitive (e.g., on some expert datasets) or even better performance (e.g., on hopper-medium-v2) against the baseline methods.

Table 1: Normalized average score comparison of SAW against baselines on D4RL MuJoCo "-v2" datasets over the final 10 evaluations and 5 seeds. r = random, m = medium, m-r = medium-replay, m-e = medium-expert, e = expert. We **bold** the first and second highest mean.

| Task Name | BC | DT | CQL | UWAC | TD3+BC | IQL | SAW (ours) |
|---|---|---|---|---|---|---|---|
| halfcheetah-r | 2.2±0.0 | - | **17.5**±1.5 | 2.3±0.0 | 11.0±1.1 | 13.1±1.3 | **23.0**±3.9 |
| hopper-r | 3.7±0.6 | - | 7.9±0.4 | 2.7±0.3 | **8.5**±0.6 | **7.9**±0.2 | 7.3±0.6 |
| walker2d-r | 1.3±0.1 | - | 5.1±1.3 | 2.0±0.4 | 1.6±1.7 | **5.4**±1.2 | **5.6**±1.5 |
| halfcheetah-m | 43.2±0.6 | 42.6±0.1 | 47.0±0.5 | 42.2±0.4 | **48.3**±0.3 | 47.4±0.2 | **47.5**±0.3 |
| hopper-m | 54.1±3.8 | 67.6±1.0 | 53.0±28.5 | 50.9±4.4 | 59.3±4.2 | **66.2**±5.7 | **95.4**±5.1 |
| walker2d-m | 70.9±11.0 | 74.0±1.4 | 73.3±17.7 | 75.4±3.0 | **83.7**±2.1 | **78.3**±8.7 | 74.8±6.9 |
| halfcheetah-m-r | 37.6±2.1 | 36.6±0.8 | **45.5**±0.7 | 35.9±3.7 | **44.6**±0.5 | 44.2±1.2 | 43.9±0.5 |
| hopper-m-r | 16.6±4.8 | 82.7±7.0 | 88.7±12.9 | 25.3±1.7 | 60.9±18.8 | **94.7**±8.6 | **97.3**±2.8 |
| walker2d-m-r | 20.3±9.8 | 66.6±3.0 | **81.8**±2.7 | 23.6±6.9 | **81.8**±5.5 | 73.8±7.1 | 58.0±6.4 |
| halfcheetah-m-e | 44.0±1.6 | 86.8±1.3 | 75.6±25.7 | 42.7±0.3 | **90.7**±4.3 | 86.7±5.3 | **89.9**±4.9 |
| hopper-m-e | 53.9±4.7 | **107.6**±1.8 | **105.6**±12.9 | 44.9±8.1 | 98.0±9.4 | 91.5±14.3 | 90.0±7.7 |
| walker2d-m-e | 90.1±13.2 | 108.1±0.2 | 107.9±1.6 | 96.5±9.1 | **110.1**±0.5 | **109.6**±1.0 | 107.2±1.9 |
| halfcheetah-e | 91.8±1.5 | - | **96.3**±1.3 | 92.9±0.6 | **96.7**±1.1 | 95.0±0.5 | 95.4±0.8 |
| hopper-e | 107.7±0.7 | - | 96.5±28.0 | **110.5**±0.5 | 107.8±7 | **109.4**±0.5 | 102.6±6.6 |
| walker2d-e | 106.7±0.2 | - | 108.5±0.5 | 108.4±0.4 | **110.2**±0.3 | **109.9**±1.2 | 103.5±6.7 |

## 4 CONCLUSION

In this paper, we explore the potential of QSS learning in offline RL. We expect the agent can reach high reward states and the action is executed to ensure that the agent can reach the desired state. We leverage *state advantage weighting* for guiding the actor and the prediction model. Our method, **S**tate **A**dvantage **W**eighting (SAW) shows good performance on D4RL datasets. To the best of our knowledge, we are the first that makes QSS learning work in offline RL. We believe our work serves as a good starting point for future studies on the state advantage.

URM STATEMENT

The authors acknowledge that at least one key author of this work meets the URM criteria of ICLR 2023 Tiny Papers Track. In this work, Aicheng Gong meets this criteria.

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

## A   RELATED WORK

**Offline RL.** Offline reinforcement learning (or batch RL) Lange et al. (2012); Levine et al. (2020) aims at learning a well-behaved policy using only the fixed dataset that was previously collected by some unknown behavior policy. A unique challenge in offline RL lies in the bootstrapping error Fujimoto et al. (2019); Kumar et al. (2019), where the agent overestimates and prefers the OOD actions, often resulting in poor performance. Current solution class involves constraining the learned policy to be close to behavior policy Fujimoto & Gu (2021); Kumar et al. (2019); Wu et al. (2019); Wang et al. (2022); Fakoor et al. (2021); Yang et al. (2022), learning completely within the dataset's support Kostrikov et al. (2022); Yang et al. (2021); Ghasemipour et al. (2021); Fujimoto et al. (2019); Zhou et al. (2020); Wang et al. (2018); Chen et al. (2020), adopting model-based methods Yu et al. (2020); Kidambi et al. (2020); Yu et al. (2021); Ovadia et al. (2019); Diehl et al. (2021), leveraging uncertainty measurement Bai et al. (2022); Wu et al. (2021); Ovadia et al. (2019); An et al. (2021), importance sampling Precup et al. (2001); Sutton et al. (2016); Liu et al. (2019); Nachum et al. (2019); Gelada & Bellemare (2019), injecting conservatism into value learning Ma et al. (2022); Kumar et al. (2020); Lyu et al. (2022); Kostrikov et al. (2021), sequential modeling Janner et al. (2021); Chen et al. (2021), etc.

**QSA and QSS Learning.**   Traditionally, the Q-learning Sutton & Barto (2018); Watkins & Dayan (1992) is conducted in the QSA style, i.e., $Q(s,a) \leftarrow Q(s,a) + \alpha\left[r + \gamma \max_{a' \in \mathcal{A}} Q(s',a') - Q(s,a)\right]$. Many algorithms are constructed based on the QSA learning, which achieve remarkable success in both discrete control Mnih et al. (2015); Bellemare et al. (2017); Wang et al. (2016) and continuous control Haarnoja et al. (2018); Fujimoto et al. (2018). Whereas, the QSS learning style is rarely studied. D3G Edwards et al. (2020) is probably the first QSS learning algorithm. D3G assumes that transition dynamics are deterministic, and shows that the QSS learning paradigm is equivalent to that of QSA learning under this assumption. Later on, there are attempts on learning from demonstration Chang et al. (2022a), improving online RL algorithms Zhang et al. (2020), planning Cyranka et al. (2022), etc. We, however, aim at making QSS succeed in offline RL. We believe our work is of great value as it may provide good insights to learning from demonstration.

**Learning from Demonstration.** Learning from observation generally refers to learning meaningful policies without access to actions Sermanet et al. (2017); Lee et al. (2021); Torabi et al. (2019b); Sun et al. (2019); Torabi (2019); Torabi et al. (2019a). It often aims at matching the performance of the expert policy Chang et al. (2022b). Unlike this setting, we adopt QSS in offline RL where the agent may encounter the non-expert datasets, rising challenges for the algorithm to learn from. Meanwhile, our SAW does not require to match the trajectories in the dataset. Instead, it selects the good next state and encourages the agent to step towards that desired state.

## B   MISSING BACKGROUND ON D3G

D3G Edwards et al. (2020) is a typical QSS learning algorithm. Note that all of the notations here are different from that in the main text. D3G has three components: the prediction model $\tau_\psi(s)$, the inverse dynamics model $I_\omega(s,s')$, and the forward dynamics model $f_\phi(s,a)$ that are parameterized by $\psi, \phi, \omega$ respectively. D3G aims at learning in a QSS style while finding the next next state $s''$ in a neighboring state set $N(s)$, i.e.,

$$Q(s,s') \leftarrow Q(s,s') + \alpha[r + \gamma \max_{s'' \in N(s')} Q(s',s'') - Q(s,s')]. \tag{1}$$

$Q(s,s')$ is undefined if $s$ and $s'$ are not neighbors. $\tau_\psi(s)$ is a function that is used to select a good neighboring state that maximizes the state $Q$-function:

$$\tau_\psi(s) = \arg\max_{s' \in N(s)} Q(s,s'). \tag{2}$$

The action is then executed to make sure that the agent can reach the proposed state, $a = I_\omega(s, \tau_\psi(s))$. D3G only estimates the $Q$-function parameterized by $\theta$. The loss function for the critic is thus given by:

$$\mathcal{L}_{\theta_i} = \mathbb{E}_{s,s'} \|Q_{\theta_i}(s,s') - r - \gamma \min_{i=1,2} Q_{\theta'_i}(s', \tau_{\psi'}(s'))\|_2^2, \tag{3}$$

where $\theta_i'$ and $\psi'$ are the target parameter, $i = 1, 2$. With the predicted next state, we update the actor (or the reverse dynamics model) via imitation learning:

$$\mathcal{L}_\omega = \mathbb{E}_{s,a,s'\sim\mathcal{D}}\|I_\omega(s, s') - a\|_2^2. \tag{4}$$

To make sure that $\tau_\psi(s)$ always proposes the neighboring state that can be reached in one step. D3G regularizes $\tau_\psi(s)$ by additionally training a forward model. The forward model is trained via:

$$\mathcal{L}_\phi = \mathbb{E}_{s,a,s'\sim\mathcal{D}}\|f_\phi(s, a) - s'\|_2^2. \tag{5}$$

D3G leverages the inverse dynamics model and the forward dynamics model to ensure the consistency of the proposed state. To be specific, given a state $s$, D3G adopts the prediction model to propose a high reward state $\hat{s}'$, and then uses the reverse dynamics model $I_\omega(s, \hat{s}')$ to determine the action that would yield that transition. Then, the action is plugged into the forward model to get the next state $s_f'$. D3G then regularizes the deviation between $s_f'$ and $\hat{s}'$. The objective function for the prediction model is then given by:

$$\mathcal{L}_\psi = -\mathbb{E}_{s\sim\mathcal{D}}[Q_\theta(s, s_f')] + \mathbb{E}_{s\sim\mathcal{D}}\|\hat{s}' - s_f'\|_2^2. \tag{6}$$

D3G relies on a strong assumption that all of the transition dynamics are deterministic and they show that QSS learning is equivalent to QSA learning under such an assumption.

## C    EXPERIMENTAL SETUP AND SAW ALGORITHM

In this section, we provide a detailed experimental setup for our proposed SAW algorithm and the baseline methods. We also include a detailed pseudo-code for the SAW algorithm.

### C.1    D4RL DATASETS

We conduct our experiment mainly on D4RL Fu et al. (2020) datasets, which are specially designed for the evaluation of offline RL algorithms. The D4RL datasets cover various dimensions that offline RL may encounter in practical applications in real world, such as passively logged data, human demonstrations, etc. The MuJoCo dataset in D4RL is collected during the interactions of different levels of policies with the continuous action environments in Gym Brockman et al. (2016) simulated by MuJoCo Todorov et al. (2012). In the main text, we evaluate our method against baselines on three tasks in this dataset, *halfcheetah, hopper, walker2d*. Each task in the MuJoCo dataset contains five types of datasets, *random, medium, medium-replay, medium-expert, expert*. **random:** a large amount of data that is collected by a random policy. **medium:** experiences collected from an early-stopped SAC policy for 1M steps. **medium-replay:** replay buffer of a policy trained up to the performance of the medium agent. **expert:** a large amount of data gathered by the SAC policy that is trained to completion. **medium-expert:** a large amount of data by mixing the medium data and expert data at a 50-50 ratio.

We adopt the normalized average score metric that is suggested in D4RL for performance evaluation of offline RL algorithms. Suppose the expected return of the random policy is $J_r$ (reference min score), and the expected return of an expert policy is $J_e$ (reference max score), the expected return of the offline RL algorithm is $J_\pi$ after training on the given dataset. Then the normalized score $\tilde{C}$ is given by Equation (7).

$$\tilde{C} = \frac{J_\pi - J_r}{J_e - J_r} \times 100. \tag{7}$$

The normalized score ranges roughly from 0 to 100, where 0 corresponds to the performance of a random policy and 100 corresponds to the performance of an expert policy. We give the detailed reference min score $J_r$ and reference max score $J_e$ for MuJoCo datasets in Table 2, where all of the tasks share the same reference min score and reference max score across different types of datasets (e.g., random, medium, etc.).

### C.2    TRAINING DETAILS OF SAW

Our method, SAW, is motivated by IQL Kostrikov et al. (2022), which learns entirely within the support of the dataset. IQL trains the value function $V(s)$ using a neural network, and leverages

Table 2: The referenced min score and max score for the MuJoCo dataset in D4RL.

| Domain | Task Name | Reference min score $J_r$ | Reference max score $J_e$ |
|--------|-----------|---------------------------|---------------------------|
| MuJoCo | halfcheetah | $-280.18$ | 12135.0 |
| MuJoCo | hopper | $-20.27$ | 3234.3 |
| MuJoCo | Walker2d | 1.63 | 4592.3 |

expectile regression for updating the critic and (action) advantage weighted regression for updating the actor. Similarly, we adopt expectile regression for the critic and (state) advantage weighted regression for updating the prediction model and the actor.

To be specific, we need to train four extra parts other than the critic, a value function $V(s)$, a forward dynamics model $F(s, a)$, a prediction model $M(s)$, and an inverse dynamics model $I(s, s')$ (the actor). The critic we want to learn is updated via expectile regression, which is closely related to the quantile regression Marilena & Domenico (2018). The expectile regression gives:

$$\arg\min_{m_\tau} \mathbb{E}_{x \sim X}[L_2^\tau(x - m_\tau)], \tag{8}$$

where $L_2^\tau(u) = |\tau - \mathbb{I}(u < 0)|u^2$, $\mathbb{I}$ is the indicator function, $X$ is a collection of some random variable. This loss generally emphasizes the contributions of $x$ values larger than $m_\tau$ and downweights those small ones. To ease the stochasticity from the environment (identical to IQL), we introduce the value function and approximate the expectile with respect to the distribution of next state, i.e.,

$$\mathcal{L}_\psi = \mathbb{E}_{s, s' \sim \mathcal{D}}[L_2^\tau(Q_{\theta'}(s, s') - V_\psi(s))], \tag{9}$$

where the state $Q$-function is parameterized by $\theta$ with a target network parameter $\theta'$, and the value function is parameterized by $\psi$. Then, the state $Q$-functions are updated with the MSE loss:

$$\mathcal{L}_\theta = \mathbb{E}_{s, s' \sim \mathcal{D}}[(r(s, s') + \gamma V_\psi(s') - Q_\theta(s, s'))^2]. \tag{10}$$

Note that in Equation (9) and (10), we only use state and next state from the fixed dataset to update the state $Q$-function and value function, leaving out any worry of bootstrapping error.

**Training the forward model.** The forward model $F_\phi(s, a)$ parameterized by $\phi$ receives the state and action as input and predicts the next state (no reward signal is predicted). A forward model is required as we want to ensure that the proposed state by our method is reachable in one step. To be specific, if we merely train one forward model $f(s)$ that predicts the next state based on the current state, there is every possibility that the proposed state is unreachable, inaccurate, or even invalid. However, if we train a forward model to predict the possible next state and encode that information in the prediction model, it can enhance the reliability of the predicted state. The forward model is trained by minimizing:

$$\mathcal{L}_\phi = \mathbb{E}_{s, a, s' \sim \mathcal{D}}\|F_\phi(s, a) - s'\|_2^2. \tag{11}$$

**Training the reverse dynamics model.** We also need the reverse dynamics model $I_\omega(s, s')$ parameterized by $\omega$ to help us identify how the agent can reach the next state $s'$ starting from the current state $s$. The inverse dynamics model is trained by weighted imitation learning, which is similar in spirit to advantage weighted regression (AWR) Peters & Schaal (2007); Kostrikov et al. (2022); Wang et al. (2018); Peng et al. (2019); Nair et al. (2020):

$$\mathcal{L}_\omega = \mathbb{E}_{s, a, s' \sim \mathcal{D}}\left[\exp\left(\beta A(s, s')\right)\|I_\omega(s, s') - a\|_2^2\right], \tag{12}$$

where $\beta \in [0, +\infty)$ is the temperature, and $A(s, s') = Q(s, s') - V(s)$ is the state advantage. By doing so, we downweight those bad actions and prefer actions that incur high reward states.

**Training the prediction model.** In the formulation of QSS, we need to evaluate the next next state $s''$. Therefore, an additional prediction network $M(s)$ is required, i.e., $s'' = M(s')$. It is critical to output a good $s''$ in QSS learning. We want that the agent can reach a high reward $s''$. To fulfill that, we maximize the value estimate on $s''$ while keeping close to the state distribution in the dataset. The prediction model $M_\xi(s)$ parameterized by $\xi$ is thus trained by minimizing:

$$\mathcal{L}_\xi = \mathbb{E}_{s, s' \sim \mathcal{D}}[\exp\left(\beta A(s, s')\right)\|s' - F_\phi(s, a')\|_2^2 - \alpha V_\psi(F_\phi(s, a'))], \tag{13}$$

where $a' = I_\omega(s, \hat{s}')$, $\hat{s}' = M_\xi(s)$. We follow Fujimoto & Gu (2021) and set $\alpha = \frac{N}{\sum_{(s_i, s_i')} |Q(s_i, s_i')|}$ across all of the tasks we evaluate. Note that all of the models we describe above are deterministic.

C.3  IMPLEMENTATION AND HYPERPAREMETERS

There are generally five components in the SAW algorithm: a value function $V_\psi(s)$ parameterized by $\psi$, a forward dynamics model $F_\phi(s, a)$ parameterized by $\phi$, double critics $Q_{\theta_i}(s, s')$ parameterized by $\theta_i, i = 1, 2$, a prediction model $M_\xi(s)$ parameterized by $\xi$, and an reverse dynamics model $I_\omega(s, s')$ parameterized by $\omega$. All of them are represented by deterministic neural networks, i.e., three-layer MLP networks. The hidden neural size is set to be 256 for all of them, and the activation function is `relu`. The learning rate for all of the learnable models is set to be $3 \times 10^{-4}$. We adopt a discount factor $\gamma = 0.99$. We list in Table 3 the detailed hyperparameters (i.e., temperature $\beta$ and expectile $\tau$) we adopt for SAW on MuJoCo tasks. We set $\beta = 5.0$ and $\tau = 0.7$ for most of the tasks. We find that our method is not sensitive to the temperature $\beta$, while the expectile $\tau$ can have comparatively larger impact on the performance of the agent. Please refer to more evidence in Appendix C.5.

The results of the baseline methods are obtained by running their official codebase, e.g., CQL (https://github.com/aviralkumar2907/CQL), UWAC (https://github.com/apple/ml-uwac), IQL (https://github.com/ikostrikov/implicit_q_learning), etc. All methods are run over 5 random seeds with their average normalized scores reported.

Table 3: The detailed hyperparameters setup for SAW on MuJoCo tasks. Normalization $\alpha = $ ✘ denotes that the value function is not normalized, and vice versa.

| Taks Name | temperature $\beta$ | expectile $\tau$ | normalization $\alpha$ |
|---|---|---|---|
| halfcheetah-random-v2 | 5.0 | 0.7 | ✘ |
| hopper-random-v2 | 5.0 | 0.7 | ✔ |
| walker2d-random-v2 | 5.0 | 0.7 | ✔ |
| halfcheetah-medium-v2 | 5.0 | 0.7 | ✔ |
| hopper-medium-v2 | 5.0 | 0.7 | ✔ |
| walker2d-medium-v2 | 5.0 | 0.7 | ✔ |
| halfcheetah-medium-replay-v2 | 5.0 | 0.7 | ✔ |
| hopper-medium-replay-v2 | 5.0 | 0.7 | ✔ |
| walker2d-medium-replay-v2 | 5.0 | 0.7 | ✔ |
| halfcheetah-medium-expert-v2 | 5.0 | 0.7 | ✔ |
| hopper-medium-expert-v2 | 5.0 | 0.3 | ✔ |
| walker2d-medium-expert-v2 | 5.0 | 0.7 | ✔ |
| halfcheetah-expert-v2 | 5.0 | 0.7 | ✔ |
| hopper-expert-v2 | 5.0 | 0.3 | ✔ |
| walker2d-expert-v2 | 5.0 | 0.7 | ✔ |

C.4  OVERALL ALGORITHM

We present the overall algorithm of SAW in Algorithm 1.

---

**Algorithm 1** State Advantage Weighting (SAW)

---

1: Initialize value network $V_\psi$, critic networks $Q_{\theta_1}, Q_{\theta_2}$, forward dynamics model $F_\phi$, prediction model $M_\xi$ and actor network $I_\omega$ with random parameters
2: Initialize target networks $\theta_1' \leftarrow \theta_1, \theta_2' \leftarrow \theta_2$ and offline replay buffer $\mathcal{D}$.
3: **for** $t = 1$ to $T$ **do**
4:     Sample a mini-batch $B = \{s, a, r, s', d\}$ from $\mathcal{D}$, where $d$ is the done flag
5:     Update value function by minimizing Equation (8)
6:     Update critics by minimizing Equation (10)
7:     Update the reverse dynamics model (actor) by minimizing Equation (12)
8:     Update the forward dynamics model by minimizing Equation (11)
9:     Update the prediction model by minimizing Equation (13)
10:     Update target networks: $\theta_i' \rightarrow \tau\theta_i + (1 - \tau)\theta_i'$, $i = 1, 2$
11: **end for**

---

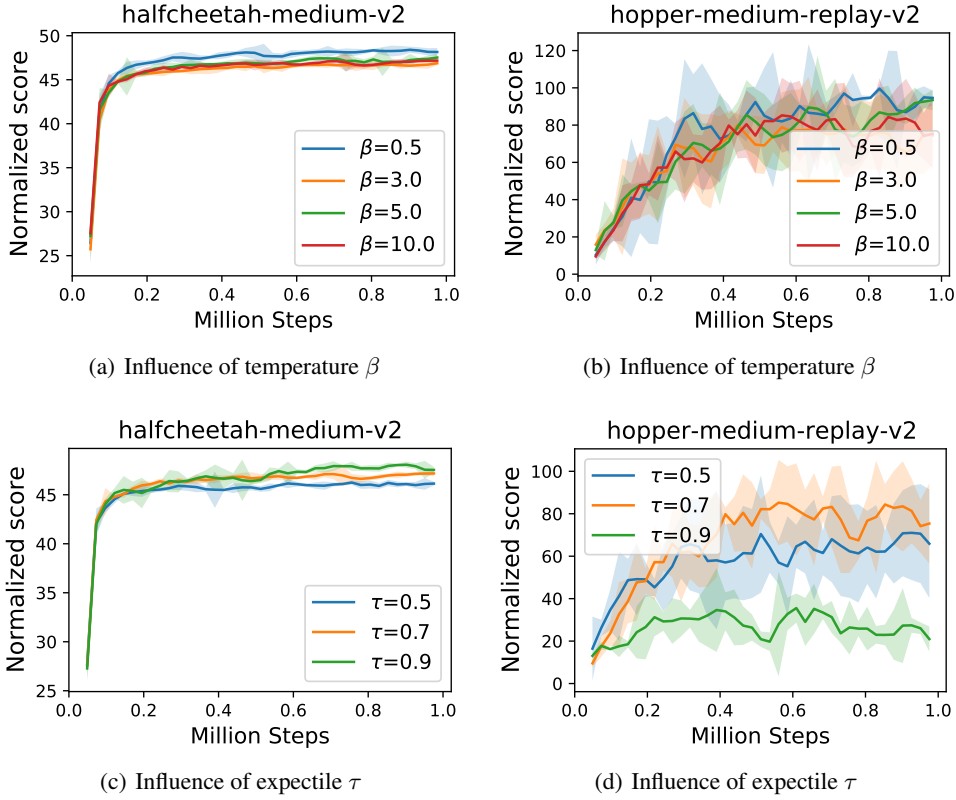

Figure 2: Parameter study of SAW on two selected datasets, halfcheetah-medium-v2 and hopper-medium-replay-v2. All experiments are run and averaged over 5 different random seeds and the shaded region denotes the standard deviation.

## C.5 PARAMETER STUDY

There are mainly two additional hyperparameters that are introduced in our SAW algorithm, the temperature $\beta$ and the expectile $\tau$. We examine the influence of these parameters by conducting experiments on two typical datasets from MuJoCo datasets, hopper-medium-replay-v2 and halfcheetah-medium-v2. We search $\beta$ among $\{0.5, 3.0, 5.0, 10.0\}$ and the expectile $\tau$ among $\{0.5, 0.7, 0.9\}$. We present the results below in Figure 2, where we find that our method is insensitive to the temperature $\beta$ (see Figure 2(a) and 2(b)), while the expectile $\tau$ can have comparatively larger impact (see Figure 2(c) and 2(d)). Smaller expectile tends to incur poor performance, while it seems that there always exists an intermediate value that can achieve the best trade-off. It is interesting that the default we adopt for these datasets may not be optimal. For example, on halfcheetah-medium-v2, $\beta = 5.0, \tau = 0.9$ exhibit better performance than $\beta = 5.0, \tau = 0.7$; on hopper-medium-replay-v2, $\beta = 0.5$ may be better. We try to unify hyperparameters to show the advantages of our method and for simplicity.

