# OpenReview forum: "State Advantage Weighting for Offline RL"
_ICLR.cc/2023/TinyPapers — Submitted to Tiny Papers @ ICLR 2023_

### Official Review · Reviewer_ny82 · 2023-03-19

**Confidence:** 4

**Summary Of Contributions:**

The paper extends a previously developed algorithm for value function approximation to be suitable for offline RL (i.e. learning from an offline dataset). In particular, they added the "state-advantage" term to the value function.

**Rating:**

Great Start (GS): a submission which meets some of the reviewing criteria but has room for improvement

**Strengths And Weaknesses:**


STRENGTHS

1. The Appendix section provides a thorough background material to help contextualize and understand the paper.
2. Extensive evaluation over different benchmarks of the MuJoCo suit and good documentation of the experiments.


WEAKNESSES
1. The paper can benefit from better writing. Both impriving the formal academic style and also the command of academic English. The narrative makes the paper hard to follow. Refer to the "suggested changes" section for more details. Some acronyms are missing explanation (like what is OOD?) and the quality of the conclussion is sub-par.

2. The algorithm shows some better performance over its counterparts, but not clear dominance over them.

3. One of the contributions of the paper is the introduction of a novel of an "Advantage" term in a well known algorithm called "QSS". In this way, the paper aims to improve QSS to be suited for offline RL. However, I don't see how the advantage term is used. I can't find it on the pseudocode or in the algorithm. The paper can be improved if the contributions are clearly listed in the introduction and the Advantage term is clearly explained on the pseudocode.

**Suggested Changes:**

1. In the Abstract, explain the acronyms before using them: QSA, QSS, OOD, IQL, SAW are used before being defined. Have in mind that your readers might not be 100% familiar with your topic, speak to a wider audience. For example, most of your readers will be familiar with RL, but not all of them with offline RL. If you write having in mind that many of us haven't seen these acronyms before, then your paper will reach wider audiences. It took me many readings to understand that actually QSA comes from Q(s,a) and QSS from Q(s, s'). Maybe even a footnote can help clarifying this aspect.

2. According to Wikipedia, English is not the first or second language by number of native speakers. However, submissions have to be done in English, which is unfortunate as academic English takes years to master. To overcome this difficulty, I would like to recommend some options to enhance the academic writing. One option is to ask on forums for peer review before submitting, some collaborator or volunteer might help you. Another source of academic writing are language models. There are currently some free online tools that will perform this task for you, i.e. if you ask a (very popular) language model: "re-write this in academic English:....", the model will give you options for conveying your idea with better grammar. Improving the readability and grammar directly impacts your chances.

3. On Section 2, you define the Q function as: "the expected discounted return by taking action a in state s". Remember to add: "under policy PI". Because for a different policy, the expected return will be different (check the definition on a textbook).

4. In the Introduction, you already cited "Edwards et al. (2020)" as a reference for D3G. You don't need to cite it afterwards.

5. The Conclusions section is subpar, meaning far away from academic writing. It is suggested to revise the Conclusions with some senior faculty or seasoned author that can provide mentoring. You can also check online on how to write a strong conclusions section for a paper. It is strongly suggested that the phrase regarding future work is re-written to match the standard practice in the field.

6. While you provide a pseudo-code for your algorithm, providing a GitHub repo will also be beneficial in the future. In this way, your paper can be adopted as a baseline for other papers. This will also help with the "reproducibility" of the paper.

7. "expectile regression" doesn't exist. You mean "quantile regression". Referring to your citation of Koenker and Hallock.

8. When you say "learning QSS in offline settings is infeasible", can you briefly explain why?

---

### Official Review · Reviewer_aPbH · 2023-04-04

**Confidence:** 3

**Summary Of Contributions:**

The paper presents a state average weighting method for offline RL. Compared with common practice that uses action advantage, it shows competitive performance (against BC, CQL, DT, UWAC, TD3+BC).

**Rating:**

High Potential (HP): a submission which meets the reviewing criteria and has potential to make an impact on the field

**Strengths And Weaknesses:**

### Strength:
1. Thorough evaluation of different domains and datasets.
2. Good sensitivity analysis on different hyperparameter choices (Sec C.5).
3. Making good use of insights from IQL.

### Weaknesses:
Please see the comments on the suggested changes.

**Suggested Changes:**

Overall, I really enjoyed this paper. I think one area of improvement is the evaluation of the paper. For example, it would be great to see some results on offline to online fine-tuning, and how it performs with different domains on D4RL. Besides, a detailed sensitivity analysis on training the prediction model would be appreciated, since paper claims that "if we train a forward model to predict the possible next state and encode that information in the prediction model, it can enhance the reliability of the predicted state".

---

### Author Response · Authors · 2023-05-30
**Opt-in for archival**

Dear reviewers, meta-reviewers and PCs,

We thank you for all of your efforts and time for this venue. We wish to opt-in for archival.

The authors.

---

> ### Comment · Area_Chair_o4aq · 2023-06-07
> **Recommend archival**
>
> After checking this paper, I confirmed that this work meets the threshold for archival, contents the URM statement and is deanonymized.

---

### Meta-Review · Area_Chair_o4aq · 2023-04-07

**Recommendation:** Invite to revise
**Confidence:** 4

**Metareview:**

The contributions of this paper are introducing a new method called state advantage weighting for offline reinforcement learning, using state advantage A(s, s′ ) and QSS learning instead of action advantage A(s, a) and QSA learning. The paper presents experiments on D4RL datasets that show the proposed method achieves remarkable performance against common baselines.

In terms of clarity, the second reviewer suggested that the paper could benefit from better writing and clearer explanations of acronyms and algorithmic contributions. The first reviewer commended the paper for its thorough evaluation and good use of insights from IQL.

Regarding originality, the paper introduces a new method that uses state advantage instead of action advantage for offline reinforcement learning, which is a novel approach in the field.

In terms of significance, the proposed method achieves remarkable performance on D4RL datasets against common baselines. This could have practical implications for applications that require offline reinforcement learning.

Overall, while the paper has strengths, such as its thorough evaluation and good use of insights from IQL, both reviewers suggest improvements, such as better writing and clearer explanations of acronyms and algorithmic contributions.



**Summary:**

a new state advantage weighting method for offline reinforcement learning and presents experiments on D4RL datasets

**Comments And Feedback To The Authors:**

Please check the valuable comments of these two reviewers and try to answer their questions/ resolve their concerns in the revised paper, especially Reviewer ny82's comments :  For example, providing additional evaluations, such as offline to online fine-tuning and more detailed sensitivity analysis on training the prediction model, as suggested by the first reviewer. Improve the writing style, provide clearer explanations of acronyms and algorithmic contributions, and revise the conclusion and future work section, as suggested by the second reviewer. Additionally, the authors can provide a GitHub repository for their algorithm to facilitate the reproducibility of their results, and clarify certain statements in the paper, such as the use of expectile regression. Overall, the authors can address the reviewers' concerns by carefully considering their feedback and incorporating changes that improve the clarity, significance, and potential impact of their work.

**Reason For Not Giving A Higher Recommendation:**

The first reviewer gave a high potential rating to this paper, which indicates that the paper meets the reviewing criteria and has the potential to make an impact on the field. However, the reviewer also suggested additional evaluations, such as offline to online fine-tuning and more detailed sensitivity analysis on training the prediction model. This suggests that the paper could benefit from further improvements and additional experimentation.

The second reviewer gave a great start rating to this paper and provided several suggestions for improvement, such as better writing, clearer explanations of acronyms and algorithmic contributions, and revisions to the conclusion and future work section. This indicates that while the paper has strengths, such as its thorough evaluation and good use of insights from IQL, there are areas that need improvement.

From my view, the paper needs extra efforts to make it more like an academical writing sample. Because scientific/academic writing is the foundation of the paper. A technique paper is supposed to be clear and understandable.

**Reason For Not Giving A Lower Recommendation:**

N/A

---

### Decision · Program_Chairs · 2023-04-10

Invite to archive